# A New Application in Biology Education: Development and Implementation of Arduino-Supported STEM Activities

**DOI:** 10.3390/biology10060506

**Published:** 2021-06-07

**Authors:** Aslı Görgülü Arı, Gülsüm Meço

**Affiliations:** Department of Science Education, Yildiz Technical University, 34220 İstanbul, Turkey; agorgulu@yildiz.edu.tr

**Keywords:** Arduino, 21st century skills/thinking skills, computational thinking, critical thinking, STEM

## Abstract

**Simple Summary:**

The rapid increase in technology in recent years has created the need to apply different methods in education. Teaching lessons with technology-based activities rather than the traditional teaching method is an obligation for teachers. For this reason, teachers need resources whose validity and reliability are tested that they can use in their lessons. In this study, a part of the technology content resource that teachers need for a biology lesson is presented. While preparing the course contents, a STEM approach including science, engineering, mathematics and technology disciplines was used. A significant increase was found in the ability of the students to whom the developed activities were applied to establish cause and effect relationships. According to this result, it can be said that Arduino-supported STEM education improves students’ abilities to establish cause and effect relationships.

**Abstract:**

Considering that generations that have grown up in the 21st-century have grown alongside technology, it is thought that integrating technology into lessons helps students learn the subject. This study aims to develop five STEM activities for the lesson of the human body systems by integrating the coding-based Arduino into STEM education. The activities were implemented to 6th-grade students for seven weeks and the effects on students’ skills of establishing a cause-effect relationship. The study method was pre-test-post-test quasi-experimental design, and the cause-effect relationship scale and semi-structured view form were used as data collection tools. As a result of the study, a significant difference was found between the Arduino-supported STEM activities developed and the students’ skills of establishing a cause-effect relationship. The students who received the Arduino-supported STEM education found the course to be entertaining and educational, and the future goals of these students were affected. In order to bring individuals who love their profession into the future, Arduino-supported STEM education should be applied and expanded in other branches and class levels.

## 1. Introduction

The word STEM is an abbreviation made up of the English initials of the words Science, Technology, Engineering, and Mathematics. STEM education, on the other hand, is an interdisciplinary approach to education that seeks solutions to real-life problems and enables the transformation of theoretical knowledge into practice and product [1]. The theoretical base of STEM education is based on the progressive educational movements and constructivism theory of the early 1900s [2]. STEM education has been defined as an effort to make all or some of the four disciplines a lesson, unit, or class based on the connections between subjects and real-life problems [3]. This effort aims to make learning connected and meaningful for students with a holistic approach that connects disciplines. In STEM education, special knowledge and skills including different practices are used. These practices are attempts to research, design, and problem-solve, together with theories, systems, and models created by scientists, mathematicians, and engineers [4]. These kinds of practices require both disciplinary knowledge and skills specific to each practice. Therefore, STEM teaching facilitates students to understand, develop and use knowledge in various practices of science, technology, engineering, and mathematics [1].

Arduino is an open source physical computing platform designed to facilitate electronic learning and programming for students and beginners. It consists of a software integrated development environment that allows users to write programs in C/C ++ programming languages that are compiled, loaded and executed on a microcontroller hardware platform [5]. Arduino is a tool that provides interaction and communication with physical parameters in daily life [6]. It is a system that has advantages such as using open source code, which is one of the biggest advantages of Arduino, having an extremely simple microprocessor circuit, and having the software package required to program the circuit with this system [7]. Arduino projects can be connected to a computer and run, or they can be run on their own. The connection of the Arduino to the computer is made through the USB interface [8]. One of the most important reasons why Arduino is popular is that it uses open-source code. In other words, it is because no code is written confidential and that these codes can be accessed easily.

Socrates’ statement “The unexamined life is not worth living” is a sentence that summarizes the importance of the ability to establish a cause-effect relationship. Questioning life and discussing what is happening within the framework of positive sciences in a cause-effect relationship can contribute to the understanding of life as a whole. When faced with a life problem, the generation raised in the 21st century is expected to strategically solve the problem and think analytically [9]. To solve the problem of daily life, students must first know the source of the problem [10], interpret it, and establish a cause and effect relationship. Therefore, providing students with the ability to establish cause-and-effect relationships is an issue that teachers should add to their agenda.

### Purpose and Significance

It is essential to use technology-supported teaching strategies according to the requirements of the 21st-century generation. Considering the studies conducted in the field of education, it is understood that the STEM approach has been adopted in recent years [11] and this approach is accepted as a modern teaching technique. For teachers to teach their lessons with the STEM approach, they must first have knowledge about this approach, adopt it, and prepare the activities they will use in their lessons. For this, teachers need a resource that consists of academically applied, valid, and reliable activities.

In this study, a validity study was conducted and the relationship between students’ skills for establishing a cause-effect relationship was examined and Arduino-supported STEM activities were developed. When the literature is examined, there is no study in which the skills of establishing a cause-effect relationship, STEM and Arduino are together.

This study aims to address the design process of Arduino-supported STEM activities that can be done for science class, human body systems lesson, and to introduce the activities, to apply the developed activities with 6th-grade students, and to determine whether the activities make a significant difference on the students’ skills for establishing a cause-effect relationship. In this context, the research seeks answers to the following questions:
(1)How can Arduino-supported STEM activities be designed for the Human Body System topic?(2)Do the developed Arduino-supported STEM activities have a significant difference in students’ skills for establishing cause-effect relationships?

## 2. Methodology

This research was conducted on students attending a public school in Istanbul in the 2020–2021 academic years. In the study, two groups, a treatment group, and a control group were determined to examine the effect of Arduino-supported STEM activities on the skills of establishing a cause-effect relationship of 6th-grade students. Since the human body systems lesson is a lesson in the 6th-grade curriculum, the students were selected from the 6th-grade.

Students were taking this lesson for the first time and they did not know about Arduino. The sample then was selected according to the easily accessible sampling method among purposeful sampling methods. While determining the treatment and control groups, the mean rank of the pre-evaluation test scores of the students was taken into consideration. Although there was no significant difference between the two groups in terms of mean rank, it was decided that the class with a low mean rank should be the treatment group. The implementation was carried out with a total of 19 students, 10 students in the treatment group and nine students in the control group. While the training was carried out by implementing the Arduino-supported STEM activities developed in the classroom representing the treatment group, the activity-based teaching method was used in the classroom representing the control group, referring to the Ministry of National Education guidebook.

### 2.1. Activity Development Process

While developing Arduino-based STEM activities, Classroom activity implementation principles [12] have been considered. Attention was paid to the purpose, use of time, classroom organization, student readiness, inclusivity, and appropriate material usage during preparing the activity. Activities begin with a knowledge-based life problem or Authentic Problems of Knowledge Society (APKS) [13]. Students learn about how to solve this problem, develop ideas, discuss their ideas for the solution of the problem with their group friends, and apply the most appropriate ideas they come up with. The necessary materials for the students to transform their ideas they find into a model are provided by the teacher. During the activity, the teacher ensures the flawless implementation of the knowledge-based life problem class activities, gives instructions to the students and the students are expected to make the most robust model at the lowest cost [14]. Teacher and student roles are shown in Table 1.

The activity stages are designed right after the teacher and student roles are determined. The activities are designed to be 3 class periods. In the first week before the activities, technical information about the Arduino is provided and the sensors and other components to be used during the activities are introduced to the students. It is aimed to motivate the students for the following weeks by stating that they will do projects with the sensors they have learned.

Lesson 1/Introduction (40 min):The teacher shortly mentions the relevant subject. Explains the achievements enriched by visuals and videos without detailing (10 min).The students are given a knowledge-based life problem for the related outcome, and the students discuss ideas for the solution of the problem among themselves. Students take on their profession and responsibilities (10 min).Information from internet-enabled computers, smartphones, and tablets to solve the problem is collected and ideas about what kind of product they will produce are developed by the students They draw the ideas they develop into the idea development notebooks; the teacher checks the drafts (20 min).

Lesson 2/Modelling (40 min):The teacher gives the students the materials they ask for. He/she puts prices on materials. Students take the materials by calculating the cost and create a model of the draft they drew in their idea development notebook (40 min).

Lesson 3/Arduino and Presentation (40 min):

The teacher shares the Arduino codes with the students. Students make the circuit connections of the sensor they will use for Arduino. They check from the computer whether the sensors are working. Students fix the Arduino to their models and give their models their final form (20 min).The students select one person among themselves as a representative. The selected person presents his/her model to the class. The teacher scores groups according to the criteria of robustness, timely completion, ability to run the Arduino, cost, and cooperative work. The group with the highest score is rewarded (20 min).

The summary of the activity schedule is provided in Table 2.

It is aimed that students acquire science, engineering, mathematics, and technology achievements simultaneously. Activity names and achievement tables for each activity are provided in Table 3.

### 2.2. Arduino Connection

In the first twenty minutes of the third lesson of the activities, students create Arduino connections. Students are expected to bring their tablets or laptop to the school. If most students do not have a tablet or laptop, the third lesson of the activities is carried out in the school’s computer laboratory. To run the Arduino, the students must install the Arduino program on their tablets or the teacher must install it on the school’s computers.

Steps of Using Arduino
Arduino software is installed on the computer.The project circuit is created on the Arduino board.The code screen opens; the codes are written. Previously created codes can also be loaded.The codes are loaded.The board executes them post-draft.It is checked whether the codes are working or not on the serial port screen.

### 2.3. Circuit Connections

A distance sensor, a thin-film pressure sensor, a dust sensor, a water flow sensor, and a moisture sensor board were used for Arduino connections. Codes were written in such a way that the distance sensor gives the warning to the LED lights of three different colors and the other sensors to the warning to the buzzer sound card. The circuit connection, including the LED lights, is shown in the skeletal and muscular system circuit connection.

For all circuits, the left wire of the buzzer sound card was connected to 5 volts, the middle wire to number 8, and the right wire to ground (GND). All circuit connections are given in Appendix A and sensor codes are given in Appendix B.

### 2.4. Validity Study for Activities: Lawshe Technique

For the scope validity of the activities prepared in this study, the Lawshe technique [15], which is based on taking expert opinions, was used. In this technique, the scope validity of the activity is expected to be determined by considering the activities developed by the researcher in terms of efficiency improvement principles and consulting to the expert opinion.

Activity development principles have been determined as the following: whether the activities are suitable for the outcome (purpose), whether the students’ prior knowledge is at a sufficient level (readiness), the time allocated to the activity is adequate (use of time), creating a classroom environment for the activity (classroom organization), mentioning the roles of teachers and students, and being suitable for the student (degree of difficulty) [11].

There are six stages of the Lawshe technique: forming a field expert group, creating a scale form, obtaining expert opinions, calculating the content validity rates for the items, determining the content validity index, and determining the items to be included in the scale by evaluating the content validity rates according to the index criteria [16]. The coverage validity rate (CVR) is calculated by dividing the number of experts who answered as “appropriate” by half of the total number of experts who gave their opinions and subtracting 1.
NG = number of experts answering “item is required/appropriate”N = Total number of experts delivering opinionsCVR= [NG/N/2]-1

According to the formula, if CVR = 1, the opinion of all experts is appropriate. If CVR = 0, the opinion of half of the experts is appropriate. CVR > 0 is more than half of the experts’ opinion is as appropriate. If CVR < 0, less than half of the experts’ opinion is appropriate.

If the number of experts is 9, the CVR rate should be a minimum of 0.75 [17]. Based on expert opinions, the CVR rate was initially calculated as 0.78. In line with the feedback from experts, the content of the circulatory system activity was changed and rearranged according to the level of the students. Also, arrangements were made in line with the feedback from experts for respiratory system activity, and the final form of the activities was given. The final version of the activities was re-presented to the experts and the CVR rate was calculated as 1.

### 2.5. Implementation of the Activities

Within the “human body systems” lesson for the 6th-grade science class within the scope of Arduino supported STEM education, five activities were implemented under the topics of the skeletal and muscular system, digestive system, circulatory system, respiratory system, and excretory system. Since 2020 was declared the year of a pandemic by the World Health Organization (WHO), formal education could not be carried out with Grade 6. Therefore, within the scope of out-of-school learning activities, 3-h activities were carried out one day a week in a STEM education center determined by the researcher, paying attention to mask, distance, and hygiene rules. The treatment group students were divided into two groups, and the study was carried out in two groups. The activities lasted 7 weeks. A cause-effect relationship scale was conducted before and at the end of the activities (Table 4).

Control group students could not be selected from the same school with the same treatment group due to pandemic conditions. In the 2020–2021 academic years, formal education was not carried out with 6th-graders, and the lessons were taught online. Therefore, in a private education course in Istanbul with the control group, the lesson on human body systems was taught face to face with the control group students. The topics were taught through traditional teaching in the control group.

### 2.6. The Models Student Designed

After the 3-period activities planned with the treatment group students for the human body systems lesson, the students developed ideas and created products for the solution of the given knowledge-based life problems (APKS).

#### 2.6.1. Skeletal and Muscular System

APKS: 3-year-old Irem injured her arm as a result of an accident. Too much movement of her arm will damage the shoulder joint. For this reason, her mother wants to wear a bracelet that measures the distance from the body that she can wear on her wrist and gives an alarm when she comes to the distance, she should not lift her arm. Let’s design the bracelet İrem needs.

For the skeletal and muscular system, the students designed a bracelet that could be worn on the wrist for people who were injured as a result of an accident and in need of physical therapy, designed a body and arm model with simple materials and connected it with the joint point. The Arduino has been mounted in a suitable spot determined by the students themselves. The students aimed to measure the appropriate distance between arm and body with a distance sensor (Figure 1).

#### 2.6.2. Digestive System

APKS: Swallowed bites getting into the trachea is a common occurrence, especially in children. To prevent this, the epiglottis valve between the trachea and stomach must work perfectly. This valve should close the trachea, especially when certain sized food pieces arrive, and the food should pass through the pharynx and enter the stomach. For this, let us design a model that includes the mouth, pharynx, stomach, trachea, and lungs. At the intersection of the trachea and esophagus, let’s develop a system that warns the epiglottis valve to close the trachea when the bite comes.

For the acquisition of this topic, students aimed to use a thin-film pressure sensor to prevent suffocation caused by particles escaping the throat, which is common in babies. The students made the connections of the pressure sensor with Arduino and designed a mouth-throat model with simple materials. They suggested placing the pressure sensor on the epiglottis cap. Thus, the bites that babies eat will alert according to their weight and the baby’s family will be able to intervene early in the congestion (Figure 2).

#### 2.6.3. Circulatory System

APKS: Mr. Sukru had to continue his life as a chronic heart patient after a heart attack. After that, he will live by paying attention to situations such as excitement, stress, and nutrition, all of which accelerate the blood flow and exhaust the heart. Therefore, taking such care should keep blood flow rate under control. For this, he needs a system that issues a warning when blood flow accelerates. Let’s help Mr. Sukru and show a system that measures blood flow on a heart model.

In the circulatory system, the students designed a pulmonary circulation model with simple materials and used transparent plastic pipes to represent vessels for the lung, heart, and heart-to-heart circulation. They used a mini water pump for blood flow. One group placed the water flow sensor in the right atrium of the heart while another group fixed it on the vein. The students developed a system that sounds an alarm when blood flow accelerates, thus aiming to intervene on time for people who have had a heart attack (Figure 3).

#### 2.6.4. Respiratory System

APKS: Hatice is a COPD patient. COPD is a chronic disease that develops due to increased sensitivity of the airways. Dust is sometimes invisible and therefore not noticed. For this, a system that informs one that dust is in the environment is needed so that COPD patients like Hatice do not enter dusty environments, ending the crisis before it begins. Let’s help Hatice by warning her about dust in the environment.

For the respiratory system, students used a sensor that measures dust content in the environment for the sensitivity of COPD (Chronic obstructive pulmonary disease) patients to dust and fixed it at the appropriate points of the nose model they designed with simple materials. One group placed the sensor under the nose, the other group placed it on the inside of the nose. It is thought that the dust sensor measures dust in the environment and stimulates when it reaches a critical level so that COPD patients will move away from that environment. (Figure 4).

#### 2.6.5. Excretory System

APKS: Kadir is sweating excessively due to his Hyperhidrosis disease. Sweating is a physiological event that occurs to control the body’s temperature. Excessive sweating (hyperhidrosis), on the other hand, is an excessive amount of sweating regardless of environmental conditions and body temperature control. Kadir should use medication according to the sweat rate on his skin. Let us develop a system that shows the pores and measures the amount of water by making a hand model for Kadir.

For this topic, the students aimed to use a moisture sensor that can be placed in the palm to inform people who have sweating problems when it is time to take medication, and which informs one when the moisture is critical. For this, they designed a hand model with simple materials and used salty water to represent the moisture. They covered the hand model with a perforated bag and provided the water flow with a syringe. Both groups found it convenient to place the moisture sensor in the palm (Figure 5). The sensor used by the students—the DTH11 moisture sensor card—is not directly operated by the arduino. For DTH11 to work, the library of this sensor has been downloaded from the Arduino website and uploaded to the Arduino.

### 2.7. Data Collection Tools

*Cause-effect relationship scale:* The cause-effect relationship scale, developed by Nuhoğlu [17]. The scale includes 10 judgment sentences about a cause-effect relationship. Students are asked to mark their thoughts about the events mentioned in these sentences according to the options “always”, “often”, “occasionally”, “rarely”, and “never”. The validity and reliability studies of the first part of the cause-effect relationship scale were conducted with 123 secondary school 6th-, 7th- and 8th-grade students. The Cronbach Alpha reliability coefficient determined for this scale consisting of 10 items, 5 of which are positive and 5 of which are negative, was found to be α = 0.88.

*Semi-Structured interview form:* In order to evaluate the activities in terms of students, a semi-structured interview form consisting of 9 open-ended questions was used for student views. Interview questions were prepared in order to reveal the advantages, disadvantages, contributions, efficiency of teamwork, attitude towards science lesson, students’ future goals and desire to prepare scientific projects for Arduino-supported STEM activities.

## 3. Results

In this section, the first research question, and the effects of the developed Arduino-supported STEM activities on students’ skills for establishing a cause-effect relationship were included.

### 3.1. Results for the Arduino-Supported STEM Activities Students Developed

The first research question is: Secondary school science class, for the “How can be designed Arduino-supported activities for the lesson of Human Body System in a science class at the secondary school level?” After the 3-period activities planned with the treatment group students for the human body systems lesson, the students developed ideas and created products for the solution of the given knowledge-based life problems.

As mentioned in the method section, 5 different models were designed by the students. These models are; a distance-sensitive bracelet model, a pressure-sensitive epiglottis model, a liquid flow-sensitive pulmonary blood circulation model, a dust-sensitive nose model, and moisture-sensitive hand model. Thus, valid and reliable Arduino-supported STEM activities that biology teachers can use while teaching human body system lesson the have been brought to the education community.

### 3.2. Results for the Effects of the Developed Arduino-Supported STEM Activities on Students’ Skills for Establishing a Cause-Effect Relationship

The second research question is: Do the STEM activities supported with Arduino have a significant contribution to the skills for establishing a cause-effect relationship of the students?

Cause-effect relationship scale was applied to the treatment and control groups to determine whether there is a significant difference between the students’ ability to establish a cause and effect relationship and the Arduino supported STEM activities developed. Research data were analyzed using SPSS 21 statistical program. Analyses were carried out using nonparametric methods due to the small number of study samples. Descriptive findings are given with mean and standard deviation values. Wilcoxon Signed Ranks test was used to compare repeated measurements. If statistically *p* < 0.05, it is accepted that there is a significant difference between the dependent variable and the independent variable [18]. The results regarding the comparison of the cause-effect relationship scale pre-test scores of the control and the treatment group students are given in Table 5.

When Table 5 is examined, it is seen that there is no significant difference between the initial means of the students. The skills for establishing a cause-effect relationship of the treatment (x^−^ = 27.00) and control group (x^−^ = 31.90) students before the study are close to each other. The results regarding the comparison of the cause-effect relationship scale the post-test scores of the control and the treatment group students are given in Table 6.

When Table 6 is examined, the mean of the treatment group students increased from 31.90 to 39.50. At the same time, there is an increase in the mean of the control group, but when the treatment group and the control group are examined together, it is seen that the mean of the treatment group students is higher than the mean of the control group. Considering that the *p*-value is 0.018 in the treatment group, it can be said that there is a significant difference between the treatment group and the control group in terms of skills for establishing a cause-effect relationship.

### 3.3. Evaluation of the Activities

In order to evaluate the activities from the perspective of the students, student views were consulted. According to the results of the semi-structured interview form filled by the students, the students evaluated the advantages and disadvantages of Arduino-supported STEM education, its contributions, counseling, group work, attitude towards science, competence, desire to prepare scientific projects and its effects on future career choices.

According to the findings obtained from the interview form, some of the students’ views are as follows:


*“After doing activities with Arduino, we learned more. For example, we understood how the arduino works and what healthcare professionals, especially doctors, do and experience.”*



*“In the activities we did with Arduino, I learned the diseases that can happen to people and learned how to develop ideas with sensors in order to find solutions.”*



*“Team work was very good. Really, our time was so good and we moved forward without ever getting bored.”*



*“I used to not like science lesson, after this activity, I started to like science lesson, it made me understand the subject better.”*



*“I used to want to be a teacher. But now I am thinking of becoming a doctor or surgeon.”*



*“In the past, I wanted to be a translator because I did not know about professions and because I was good at foregin language, now I want to be a scientist or professor.”*


According to the opinions of the students about the activities;
✓they understood the subject better,✓they have decided to choose a profession in the field of health in the future,✓found the activities fun and efficient,✓they liked science lesson more.

## 4. Discussion

In this study, Arduino-supported STEM activities were developed, the developed activities were implemented to middle school students, and the effects of the activities on students’ ability to establish cause and effect relationships were examined.

According to the results of the study, the students developed different projects for the five topics of the human body systems lesson. The students prepared projects by integrating biology subjects with electronics and robotics for the solution of the given information-based life problem. Since Arduino includes a simple circuit connection [19], teachers think that Arduino is more suitable for physics subjects and they designed and implemented activities in the physics subjects of the science class [20,21]. When the studies done with Arduino are examined, it is seen that biology is given a very little place in the studies [22,23]. With this study, biology was combined with physics and robotics and a resource that teachers can use in their lessons related to these three fields is presented.

Students developed different ideas for the solution to the information-based life problem given during the activities. Different students designed different models to solve the same problem and fixed the Arduino in different places. Unlike traditional education, STEM education is an education method where students are at the center, work in groups, and develop ideas [24]. The use of technology in education is a necessity for the twenty-first century generation that grew up with technology [25]. For this reason, this study will shed light on future studies, as it is a study that combines STEM education with technology.

When the studies conducted in the field of education in recent years are examined, the significance of the skills of the twenty-first century is mentioned [26] and many skills such as critical thinking [27], problem-solving skills [28,29], working in groups [30], analysis-synthesis [31] are tried to be gained to students. For students to be able to solve problems, they must first understand the cause of the problem and predict the results of the solution they found [32]. In this study, it is seen that there is a significant difference between students’ skills for establishing cause-effect relationships and Arduino-supported STEM education. Therefore, it can be said that Arduino-supported STEM education gives students the skill for establishing cause-effect relationships. Studies have shown that STEM education increases the academic achievement of students [33] changes their interest in the course positively [34], gives them critical thinking [35], and increases their desire to prepare scientific projects. This study proved that STEM education gives students the skill for establishing cause-effect relationships.

## 5. Conclusions

Considering the contributions of the Arduino-supported STEM study to students, teachers can apply such activities in other subjects of the science lesson, do similar studies for other gains of the systems unit in our body, and apply them with all grade levels. In addition, Considering the effects of the Arduino-supported STEM study on the future goals of the students, the Ministry of National Education can purchase and supply arduino for schools, train teachers in coding with arduino, and the work can be expanded throughout the country in order to train doctors or engineers who love their profession for the future.

## Figures and Tables

**Figure 1 biology-10-00506-f001:**
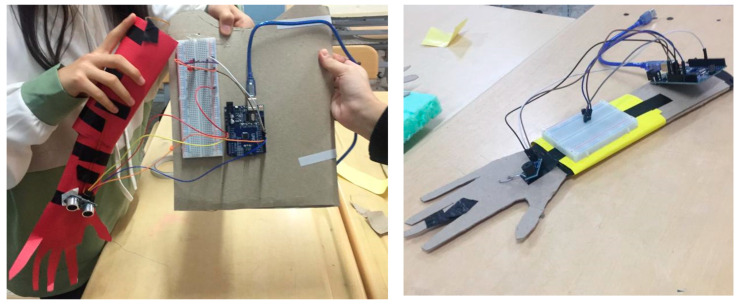
Bracelet model with a distance sensor.

**Figure 2 biology-10-00506-f002:**
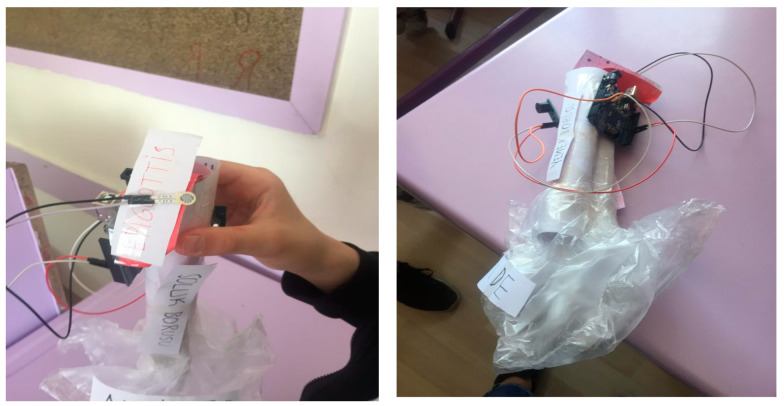
Epiglottis model with a thin film pressure sensor.

**Figure 3 biology-10-00506-f003:**
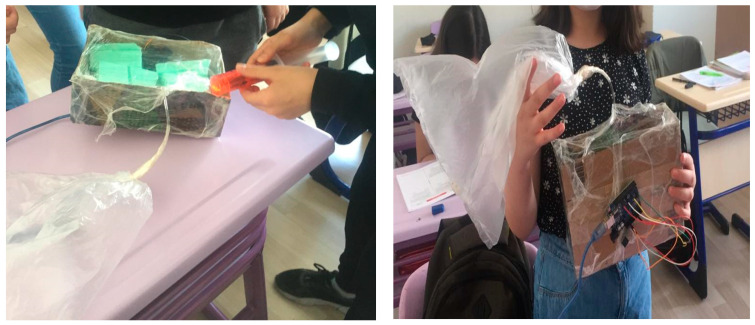
Pulmonary Circulation with a waterflow sensor. The plastic bag represents the lung and the box represents the heart.

**Figure 4 biology-10-00506-f004:**
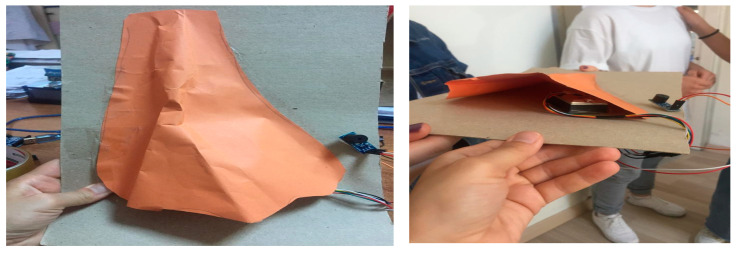
Nose model with a dust sensor.

**Figure 5 biology-10-00506-f005:**
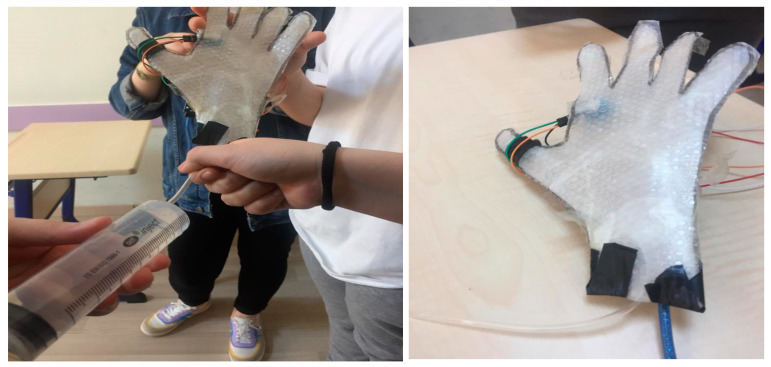
Hand model with a moisture sensor.

**Table 1 biology-10-00506-t001:** Teacher and Student Roles.

Teacher Role	Student Role
(1) Gives technical information about Arduino, introduces the sensors. (2) Presents the knowledge-based life problem. (3) Encourages students to practice the activity steps. (4) Listens to the ideas the students find. (5) Checks the students’ idea drawings. (6) Checks the robustness of prototypes. (7) Evaluates the groups, scores them.	(1) Learn about the solution to the problem. (2) Works as a team. (3) Undertakes the most appropriate task within the team. (4) Brainstorm with friends and develops ideas for the solution to the problem. (5) Drafts the idea he/she finds (6) Makes the product prototype with the given materials. (7) Presents its product

**Table 2 biology-10-00506-t002:** The Activity Schedule.

Duration (min.)	Lesson 1	Lesson 2	Lesson 3
10	Obtaining Information	Modeling	Arduino Connection
10	Task Distribution		
10	Developing Ideas		Presentation
10			

**Table 3 biology-10-00506-t003:** Activity Achievements.

Subject	Science	Engineering	Mathematic	Technology	Social Product
Skeletal and muscular system (distance-sensitive bracelet model)	Explains the structures of the skeletal and muscular system with examples.	Identifies the processes involved in an engineering project. Explains the stages such as planning, prototyping, design, execution, quality control, and reporting. Predicts the performance, reliability, and failure status of alternative solutions. Investigates the principles and elements of design.	Compares and orders objects by length Selects and measures the appropriate non-standard measuring tool to measure a length Creates and draws structures by using shapes and models. Measures lengths in meters or centimeters using standard tools Draws a segment with a given length using a ruler.	Realizes that computers can be used for different purposes. Uses information technology tools to do research. Does simple research on the Internet Collects data about a problem Understands data collection with Arduino and how to use sensors.	Communicates effectively with groupmates and shares ideas Can transform his/her imagination into a drawing Participates actively in group work The student presents the designed product to the class in an intelligible manner.
Digestive system (pressure-sensitive epiglottis model)	Explains the functions of the structures and organs that make up the digestive system using models.				
The circulatory system (pulmonary blood circulation model)	Explains the functions of structures and organs that make up the circulatory system using a model.				
The respiratory system (dust-sensitive nose model)	Explains the functions of the structures and organs that make up the respiratory system using models.				
Excretory system (moisture-sensitive hand model)	Summarizes their functions by showing the structures and organs that make up the excretory system on models.				

**Table 4 biology-10-00506-t004:** Activity Implementation Schedule.

Time	Activity
1st week	Cause-Effect Relationship Scale pre-test implementationv Introduction of Arduino and sensors to be used in events
2nd week	Skeletal and muscular system (Distance-sensitive bracelet model)
3rd week	Digestive system (Pressure-sensitive epiglottis model)
4th week	The circulatory system (Liquid flow-sensitive pulmonary blood circulation model)
5th week	The respiratory system (Dust-sensitive nose model)
6th week	Excretory system (Moisture-sensitive hand model)
7th week	General review of human body systems subject Cause-Effect Relationship Scale post-test application

**Table 5 biology-10-00506-t005:** Results Regarding the Comparison of Cause-Effect Relationship Scale *Pre-Test Scores* of the Control and Treatment Group Students.

Groups	*N*	x^−^	S	U	*p*
Control Group	9	27.00	1.05	43.500	0.069
Treatment Group	10	31.90	5.48		

**Table 6 biology-10-00506-t006:** Results Regarding *Pre-Test-Post-Test* Comparative Cause-Result Relationship Scale of Control and Treatment Group Students.

Groups	*N*	Pre-Test		Post-Test		Z	*p*
		x^−^	SS	x^−^	SS		
Control Group	9	27.00	6.38	31.11	1.05	−2.675	0.069
Treatment Group	10	31.90	4.38	39.50	5.48	−2.374	**0.018 ***

* *p* < 0.05.

## Data Availability

The data that support the findings of this study are available from the corresponding author upon reasonable request.

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
