# Peer review of "A New Application in Biology Education: Development and Implementation of Arduino-Supported STEM Activities"

_biology, 2021, doi:10.3390/biology10060506_

Round 1

Reviewer 1 Report

Dear authors, 

The aim of integrating Arduino into biological education to motivate students is very interesting. However, I have some issues for you:

  1. In my experience, 20 minutes to introduce Arduino hardware, programming and sensors is not enough.
  2. Was there any kind of evaluation of the activities? If yes, please comment on how it was and the results.
  3. I think, "robotic" is out of the content of this article. Please remove this word in the article.
  4. Did you have any feedback from students?
  5. Activities 3.1.1 to 3.1.5 should include more details (photos, models, etc). 
  6. Tables 11 and 12 don´t exist.
  7. In Tables 5 and 6, please change "," by "."
  8. Perhaps, lines 386 to 398 should be better in the introduction section. References should not be in the conclusion section.

Reviewer 2 Report

First sentence in abstract is awkward and cold be deleted

Line 33 is awkward

Line 34 what practices – choppy sentences are not flowing together – do you mean STEM practices?

Line 39 makes me so angry I would have stopped reading if I was not reviewing. The main emphasis for STEM and the stem program I have ran for 20 years is not technology!!! I would agree with line 41

Before line 43 explain what Arduino is

Line 52-58 I really like what you are trying to do in the last paragraph of the introduction but the sentence flow is awkward – I would look to rewrite

Line 79 makes no sense  ‘How can be designed’ what does this mean – the rest of the sentences is awkward as well

Line 83 period in front of methodology

Line 84 delete first sentence – not useful

Line 84 – 90 this whole first paragraph is not useful. Just tell the reader what you did. I would delete this whole paragraph. The next paragraph says all that needs to be said

Line 102 remove 6th graders – you just said it

Line 108 awkward – you keep repeating 6 grade – the reader gets it you are using 6th graders

Line 110 – 112 what does this mean – two sentences that do not say anything – how about We chose ……… to sample the data in the class. One simple sentence

Line 123 no teaching is flawless

Is table 1 useful and why. I do not understand it’s importance

Line 185 what is GND

Line 227 groups of 5 with 2 student per group

Also you had one class with 9 how did they deal with that

Did the two groups have the same teacher – if not then you findings could be just better teacher and nothing to do with treatment

Line 254-256 remove not useful

Line 258 question does not make sense

3.1 is this results or methods I think this is in the wrong section. I like it but not in results because it is not a result. It is what the student scenarios are for the class.

Line 336-338 awkward sentence – to many ands as well

336-343 confusing even looking back at methods I do not understand fully what you compared – this needs clarification

Line 345-346 tell me what is in the table, don’t tell me to look at table.

Line 349-351 poorly written

Overall, the method section and results is poorly written creating a lot of confusion

The discussion is written better but with the lack of understanding from results and methods, I am not sure I trust the outcomes.

Round 2

Reviewer 2 Report

I think the authors have made the changes needed. I still have a few issues with the writing but some of this is style.